# Lactoferrin Induces Erythropoietin Synthesis and Rescues Cognitive Functions in the Offspring of Rats Subjected to Prenatal Hypoxia

**DOI:** 10.3390/nu14071399

**Published:** 2022-03-27

**Authors:** Alexey V. Sokolov, Nadezhda M. Dubrovskaya, Valeria A. Kostevich, Dmitrii S. Vasilev, Irina V. Voynova, Elena T. Zakharova, Olga L. Runova, Igor V. Semak, Alexander I. Budevich, Natalia N. Nalivaeva, Vadim B. Vasilyev

**Affiliations:** 1Department of Molecular Genetics, Institute of Experimental Medicine, Acad. Pavlov Str. 12, 197376 Saint-Petersburg, Russia; hfa-2005@yandex.ru (V.A.K.); iravoynova@mail.ru (I.V.V.); et_zakharova@mail.ru (E.T.Z.); o.runova@yandex.ru (O.L.R.); 2Faculty of Dental Medicine and Medical Technologies, Saint Petersburg State University, 8A 21st Line V.O., 199034 Saint-Petersburg, Russia; 3Laboratory of Physiology and Pathology of CNS, Sechenov Institute of Evolutionary Physiology and Biochemistry, 44 Thorez Ave., 194223 Saint-Petersburg, Russia; ndub@mail.ru (N.M.D.); dvasilyev@bk.ru (D.S.V.); natalia.nalivaeva@outlook.com (N.N.N.); 4Department of Biochemistry, Faculty of Biology, Belarusian State University, Nezavisimisty Ave. 4, 220030 Minsk, Belarus; semak@bsu.by; 5Scientific and Practical Centre on Animal Husbandry of the National Academy of Sciences of Belarus, 11 Frunze Str., 222160 Zhodino, Belarus; budevich7388100@mail.ru

**Keywords:** hypoxia, pregnancy, cognitive functions, lactoferrin, hypoxia-inducible factor, erythropoietin

## Abstract

The protective effects of recombinant human lactoferrin rhLF (branded “CAPRABEL™”) on the cognitive functions of rat offspring subjected to prenatal hypoxia (7% O_2_, 3 h, 14th day of gestation) have been analyzed. About 90% of rhLF in CAPRABEL was iron-free (apo-LF). Rat dams received several injections of 10 mg of CAPRABEL during either gestation (before and after the hypoxic attack) or lactation. Western blotting revealed the appearance of erythropoietin (EPO) alongside the hypoxia-inducible factors (HIFs) in organ homogenates of apo-rhLF-treated pregnant females, their embryos (but not placentas), and in suckling pups from the dams treated with apo-rhLF during lactation. Apo-rhLF injected to rat dams either during pregnancy or nurturing the pups was able to rescue cognitive deficits caused by prenatal hypoxia and improve various types of memory both in young and adult offspring when tested in the radial maze and by the Novel Object Recognition (NOR) test. The data obtained suggested that the apo-form of human LF injected to female rats during gestation or lactation protects the cognitive functions of their offspring impaired by prenatal hypoxia.

## 1. Introduction

Oxygen deficiency in the embryonic period results in a large spectrum of postnatal behavioral alterations and cognitive dysfunctions [1]. The periods of increased sensitivity of the brain to deleterious factors reflect the heterochronous ontogenetic development of the nervous system [2]. Detrimental effects in those critical periods of prenatal development can cause structure-functional perturbations at every level of cerebral organization. In particular, the impaired development of the brain cortex and hippocampus, which play the key role in learning and memory, can result in cognitive deficiency.

As shown in our studies in rats, prenatal hypoxia on embryonic days E13–14, when the generation and migration of the cells into the forming cortical areas take place [3], leads to delayed development of pups’ motor reactions and long-lasting cognitive dysfunctions observed at various stages of ontogenesis and underlined by impaired synaptic plasticity [4,5,6]. Along with the functional and structural alterations caused in animals by prenatal hypoxia, substantial rearrangements at the molecular and metabolic levels were observed in their nervous system. Firstly, this was manifested by the altered expression of various neuronal genes, resulting in the changes in mRNAs and protein levels of various enzymes and leading to malfunctioning of signal transduction systems, including cholinergic and purinergic mediation [7,8]. Understanding the molecular and cellular processes that underlie cognitive impairment is essential for monitoring the development of an organism and for designing the corrective interventions to preserve its vital functions. In recent years, lactoferrin (LF) was shown to possess some protective functions which suggested that it may be beneficial against the effects of prenatal hypoxia.

Lactoferrin (LF) is a cationic transferrin found in milk, some other exocrine secretions, and in neutrophils. It has been shown to possess antibacterial, antifungal, antiviral and antiparasitic, anti-inflammatory, immunomodulatory and other activities [9,10,11,12,13,14,15,16,17]. Almost 90% of LF in human milk is iron-free (apo-LF) and has an extreme affinity toward Fe(III), which makes it an efficient iron chelator [18].

A bulk of evidence has been accumulated concerning the beneficial effect of breast feeding on brain development in neonates, especially in preterm babies [19]. The importance of LF in breast milk for neurodevelopment and cognition was emphasized in a comprehensive review [20].

We showed recently that at normoxia, the inactivation of hypoxia-inducible factors (HIF-1α and HIF-2α) is prevented by apo-LF [21,22]. Our data were supported by the finding of another research group showing that HIF is upregulated in preterm piglets treated with bovine LF [23]. This endows LF with the capacity to protect an organism against experimental hypoxia and anemia. In the absence of apo-LF, as iron chelator, this effect is not observed, since hydroxylated HIFs undergo ubiquitination and proteasomal degradation. In case of hypoxia or iron deficiency, HIF-1α and HIF-2α are not hydroxylated but make their way towards the nucleus to stimulate expression of about 200 genes, half of which secure the survival of a cell under stress.

Our studies demonstrated that various tissues of animals treated with recombinant human LF (rhLF) responded by expressing HIF-1α and HIF-2α target genes [22,24]. As a result, such proteins as erythropoietin (EPO) and ceruloplasmin, which both are products of HIF-2α and HIF-1α target genes, were synthesized in noticeable amounts [22]. These data allowed us to suggest that LF is an efficient natural anti-hypoxic agent. The aim of the present study was to investigate whether rhLF administration during gestation or lactation to rats exposed to hypoxia at pregnancy might mitigate or abrogate cognitive defects in their offspring.

## 2. Materials and Methods

### 2.1. Reagents

Recombinant human apo-lactoferrin (rhLF) purified from the milk of transgenic goats was obtained at the Belarusian State University and RUE «Scientific and Practical Centre of the National Academy of Sciences of Belarus for Animal Production». The product is officially branded “CAPRABEL™” and contains about 90% of the iron-free form of LF (apo-LF) [25]. Monoclonal mouse antibodies against recombinant human EPO (A 171 PC/Unit 7, “Protein Contour Ltd.”, Saint Petersburg, Russia); monoclonal mouse antibodies against HIF-1α (clone H1a1pha67) and HIF-2α [ep 190b] («Abcam», Cambridge, UK); rabbit antibodies against mouse IgG; and dimethyl sulfoxide were all from Sigma (USA); horseradish peroxidase-labeled goat antibodies against rabbit immunoglobulins and dry skimmed milk were all from Bio-Rad (Hercules, CA, USA); acrylamide, N,N′ -methylenebis(acrylamide), N,N,N′,N′—tetramethylethylene diamine («Medigen», Novosibirsk, Russia); ammonium persulfate, glycerol, Coomassie R-250, mercaptoethanol, sucrose and Tris were all from «Serva» (Germany); PBS (phosphate buffer saline)—0.15 M NaCl, pH 7.4, 1.9 mM Na_2_HPO_4_/8.1 mM NaH_2_PO_4_. Rabbit polyclonal affinity antibody against human LF was obtained in house as described earlier [26].

### 2.2. Animals

Wistar rats bred in the vivarium of the I.M. Sechenov Institute of Evolutionary Physiology and Biochemistry of the Russian Academy of Sciences (St. Petersburg, Russia) were used. All experimental protocols, animal housing and tissue collection have been carried out in accordance with the ARRIVEguidelines for work with experimental animals [27] and the instructions of the Russian Academy of Sciences. Pregnant rats and their offspring were kept under 12-h light/dark cycle and with free access to food and water. The study was carried out in pregnant and lactating rats and their embryos, as well as in males from the offspring of females of the control and three experimental groups: prenatal hypoxia on E14 with saline injections; prenatal hypoxia on E14 with apo-rhLF treatment during pregnancy on E9, E11, E13, E15; and prenatal hypoxia with apo-rhLF treatment during nurturing on P0-P15.

Pregnant Wistar rats (200 g) were subjected to hypoxia on the 14th day of gestation (E14) in a special chamber as described below. Half the pregnant rats were treated by apo-rhLF, (*i.p.* injections of 10 mg CAPRABEL dissolved in 0.5 mL saline per rat) on the 9th, 11th, 13th, and 15th days of gestation (E9, E11, E13, E15) or during nurturing (daily after delivery on P0 up to P15) and were sacrificed for biochemical experiments on E14 and P14, respectively (Figure 1). 

Organ homogenates of females, embryos and suckling pups were analyzed by Western blotting with anti-HIFs or anti-EPO. Another group of pups was allowed to grow up, and their memory tests were performed starting from postnatal days 22 (P22) or 90 (P90).

Table 1 shows the distribution of animals among groups and encompasses the variety of experiments accomplished.

### 2.3. Model of Prenatal Normobaric Hypoxia

On the 14th day of gestation, 2 groups of pregnant Wistar rats (200 g)—treated with saline or with apo-hrLF on E9, E11 and E13—were exposed to normobaric hypoxia in a 100 L chamber equipped with systems for ventilation, thermoregulation, adsorption of exhaled CO_2_ and gas analysis. To create hypoxic conditions, the oxygen level in the chamber was gradually reduced from 20.7 to 7.0% and maintained at this level for 3 h. CO_2_ concentration inside the chamber was kept below 0.2%, and the temperature was 22 °C. Control rats were kept for the duration of the experiment in the same room under normal oxygen content. Some pregnant rats and embryos from the control and experimental groups were taken for biochemical experiments 4 h after the end of hypoxic insult. The remaining pregnant rats were placed in individual cages and injected with saline or apo-rhLF on E15. On the 2nd day after birth, only 8 pups were left in each brood. When calculating the age of rat pups, the day of their birth was considered as P0.

### 2.4. Collection of Milk Samples

To analyze the presence of rhLF in milk 3 h after a single injection of apo-rhLF (10 mg in 0.5 mL of saline), intact lactating rats (*n* = 3) were separated from the pups (P14) for an hour and given an intraperitoneal injection of oxytocin (10 IU/kg body mass). After 10 min, rats were etherized, and ca. 1 mL of milk was obtained from each animal using a special peristaltic pump with a soft silicon tip on the tube. Samples of whole milk were centrifuged, and the supernatants collected for the analysis of rhLF by Western blotting with rabbit IgG against human LF. The presence of rhLF was also determined in gastric content of some nurtured pups selected at random (*n* = 6), which were etherized and decapitated.

### 2.5. Western Blotting

Animal tissues/organs were homogenized on ice and lysed in 3-fold volume (*v/w*) of a buffer (250 mM sucrose, 25 mM NaCl, 2 mM EDTA, 50 mM Tris-HCl, pH 7.4), containing a cocktail of proteinase inhibitors. Lysates were centrifuged for 15 min at 15,000 g (4 °C). The content of proteins in the supernatants was assayed according to Bradford’s method [28]. The samples containing 50 µg of protein in loading buffer were boiled for 10 min and applied on each lane of SDS-PAGE block for electrophoresis [29]. After electrophoresis, proteins were transferred during 1 h to a nitrocellulose membrane by semi-dry electric transfer [30]. The nitrocellulose membrane was then soaked for 30 min in BLOTTO-T (3% solution of dry skimmed milk in PBS containing 0.05% Tween 20) and sequentially incubated for 12 h at 4 °C in BLOTTO-T containing murine monoclonal antibodies (a) to EPO (1:5000), (b) to HIF-1-α (1:1000), (c) to HIF-2-α (1:1000) or (d) to LF (1:5000). After 3 washings in BLOTTO-T, the nitrocellulose membranes were incubated for 1 h at 37 °C with a secondary rabbit antibody against mouse IgG (except anti-LF) and then with HRP-labelled goat antibodies against rabbit immunoglobulins (1:1000). Immunoreactive bands were revealed using the following chromogenic mixture: 6 mg of 4-chloro-1-naphthol in 2 mL of methanol with 10 mL of 5 mM H_2_O_2_ in PBS. The developed violet bands were scanned, and the nitrocellulose membranes were further kept in the darkness.

### 2.6. Novel Object Recognition Test (NOR Test)

Assessment of the short- (STM) and long-term memory (LTM) by the novel object recognition test (NOR) [31,32] was performed as described previously [6]. At the beginning of the test, each animal was adapted for 5 min to a 100 × 100 cm box with 20 mm high non-transparent walls in the absence of any specific stimuli. During the first training session, following 2 h of acclimatization in the experimental area, two objects (number 1 and 2) were placed and left for the animal to explore them for 10 min. The test was repeated after 10 min to assess STM and then after 60 min and 24 h later to assess LTM. During each consecutive test, which lasted 5 min, one of the objects (object 2) was replaced by a new object (numbered 3 for STM testing, and 4 and 5 for testing LTM) while object 1 stayed unchanged the whole time. The time (in sec) during which the animals explored each object was recorded by an observer unaware of the experimental group and expressed in % of the total exploration time of both objects. The same scheme of experiments was used for testing the offspring of both ages (young—P22 and adult—P90) in each experimental group: control rats (*n* = 32), rats subjected to prenatal hypoxia (*n* = 32), rats subjected to prenatal hypoxia and treated with apo-rhLF during gestation (*n* = 22) and lactation (*n* = 24).

### 2.7. Working Memory Analysis in the Eight-Arm Radial Maze

Working memory of rats from all experimental groups was tested at the age of 3 months (P90) in the 8-arm radial maze as described in detail previously [33]. For this, following 24 h food deprivation, the rat was placed on a platform in the 2-level, 8-arm maze from where they could enter any of the maze arms. Each arm had a food pellet (35 mg, made of sugar and starch). The rat was allowed to perform only 8 visits of the arms, and any visit to the same arm was counted as an error. Working memory was evaluated and expressed as % of errors from 8 visits. The same scheme of experiments was employed for testing offspring of both ages in each experimental group: control rats (*n* = 7), rats subjected to prenatal hypoxia (*n* = 7), rats subjected to prenatal hypoxia and treated with rhLF during gestation (*n* = 7) and lactation (*n* = 7).

### 2.8. Statistical Analysis

All statistical tests were performed using Statistica 8.0 software (Statsoft Inc., Tulsa, OK, USA), SPSS 17.0 for Windows and GraphPad Prism 9.2.0 (San Diego, CA, USA). In the analysis of animal behavior, we used one sample *t*-test to identify the differences between the time of recognizing a novel object and the hypothetical 50% value which testifies to the equivalent preference of the novel and familiar objects and 1-way ANOVA with the Tukey–Kramer *post hoc* test for the radial maze test. An α level of 0.05 was used as a significance criterion for all tests. The data are given as mean ± standard error of mean. The two-tailed Fisher exact test was used in the analysis of the EPO and HIF-1α and HIF-2α presence.

## 3. Results

### 3.1. Lactoferrin-Induced Upregulation of EPO in Pregnant Rats and Their Pups

Immunoblotting analysis of the brains of pregnant rats and the brains and torsos of their embryos on E14 has revealed that in the animals of saline-treated control group and saline-treated group submitted to hypoxia there were no detectable amounts of EPO. Hypoxia on E14 has not affected the levels of EPO in all organs studied. However, in the group of rats which were subjected to hypoxia and received injections of apo-rhLF, there was a significant increase in EPO protein levels in the brain both of pregnant rats and their embryos, as well as in the fetal bodies but not in the placentas (Figure 2, Table 2, representative Western blotting images—Appendix A). 

The number of pregnant females and embryos with detectable or undetectable EPO levels was analyzed by Fisher exact test. It confirms that the injection of apo-rhLF leads to a statistically significant increase in EPO in the brain tissue of embryos from pregnant females subjected to hypoxia (Fisher exact test, *p* < 0.05), but prenatal hypoxia itself did not significantly affect the content of EPO in the brain tissue of embryos (Fisher exact test, *p* > 0.05).

The increased synthesis of EPO in the pregnant rat brain and in the brain and bodies of their embryos correlates with apo-rhLF administration, since, previously, we have shown that CAPRABEL stabilizes the levels of HIF-1α and HIF-2α in the same organs of naïve pregnant rats and their embryos [21,24].

In a separate experiment, we have analyzed whether the injection of apo-rhLF to lactating dams resulted in the appearance of human apo-LF in their milk or in the gastric content of the pups (Figure 3). It was found that a single injection of human apo-LF led to the appearance of this protein in the milk of lactating dams and its persistence there for 4 to 24 h. It was also detected in the gastric content of the pups.

The analysis of the EPO, HIF-1α and HIF-2α levels in the pup (P14) brain, liver and spleen by Western blotting has revealed that they were detectable in these organs only in rats whose mothers received apo-rhLF (Table 3). 

The number of pups with detectable or undetectable EPO levels was analyzed by the Fisher exact test, which confirmed that the injection of apo-rhLF during lactation led to the increase in EPO in the brain tissue of pups subjected to prenatal hypoxia (Table 2, Fisher exact test, *p* < 0.05). However, prenatal hypoxia itself did not significantly affect the EPO content in the brain of pups whose mothers have not been treated with apo-rhLF (Fisher exact test, *p* > 0.05).

These data strongly suggest that the treatment of pregnant rats with apo-rhLF results in the upregulation of the EPO protein both in their brain and the brains and bodies of their embryos. The injection of apo-rhLF to lactating rats led to the appearance of rhLF in their milk and the gastric content of their pups, which was accompanied by the upregulation of HIF-1α, HIF-2α and EPO in the brain, liver and spleen of the pups.

### 3.2. Effect of apo-rhLF Injections to Pregnant and Lactating Rats on the Memory of Their Offspring

The results of testing rat pups, whose mothers were subjected to hypoxia and treated with saline injections have demonstrated that both on P22 and P90 there was statistically significant impairment of their short- and long-term memory compared with control rats when examined by NOR and the two-level, eight-arm radial maze (Figure 4 and Figure 5).

Normally, intact rats prefer to spend more time exploring novel rather than familiar objects for 10 min, 60 min or 24 h intervals after training, which is indicative of normal short-term (STM) and long-term memory (LTM1 and LTM2) (Figure 4). In fact, in the offspring of rats not subjected to hypoxia during pregnancy, the time spent to recognize a novel object differed significantly from the hypothetical 50% level in the testing session performed 10 min after the first training session (STM, one-sample *t*-test: *t**_young_* = 3.061, *p* = 0.007 and *t**_adult_* = 3.554, *p* = 0.003), after 60 min (LTM1, *t**_young_* = 2.609, *p* = 0.019 and *t**_adult_* = 5.152, *p* < 0.001) or 24 h after the first training session (LTM2, *t**_young_* = 4.476, *p* < 0.001 and *t**_adult_* = 4.459, *p* = 0.001). On the contrary, rat pups subjected to prenatal hypoxia had no preference (*p* > 0.05) for the novel object but spent about 50% of the time exploring both objects (Figure 4).

Young (P22) and adult rats (P90) born from the mothers that experienced hypoxia and treatment with apo-rhLF during pregnancy showed the results in the test for the STM and LTM, similar to the control pups, as their recognition of the novel object significantly differed from the hypothetical 50%: 10 min after training (one-sample *t*-test: *t**_young_* = 2.487, *p* = 0.026 and *t**_adult_* = 4.523, *p* = 0.004); after 60 min (*t**_young_* = 7.643, *p* <0.001 and *t**_adult_* = 4.439, *p* = 0.004); and after 24 h (*t**_young_* = 6.250, *p* < 0.001 and *t**_adult_* = 4.797, *p* = 0.003). Moreover, young and adult rats born from the mothers subjected to hypoxia and treated with apo-rhLF during lactation had normal (like the control group) STM and LTM1 on P22 (one-sample *t*-test: *t**_young_* = 5.437, *p* <0.001 and *t**_young_* = 7.235, *p* <0.001, correspondingly) and normal STM, LTM1 and LTM2 on P90 (one-sample *t*-test: *t**_adult_* = 2.923, *p* = 0.027, *t**_adult_* = 3.510, *p* =0.013 and *t**_adult_* = 2.856, *p* = 0.029).

The analysis of rat behavior in the two-level eight-arm radial maze showed that the number of erroneous runs in adult rats (P90) subjected to prenatal hypoxia increased by 3.6 times compared with the control rats (Figure 5, 1-way ANOVA with the Tukey–Kramer post hoc test, *q* = 6010, *p* =0.002). The treatment of pregnant rats with apo-rhLF during pregnancy prevented memory deterioration in their offspring, which resulted in the 2.9-fold decrease in the number of errors (one-way ANOVA with the Tukey–Kramer *post hoc* test, *q* = 5602, *p* = 0.003) (Figure 5), making them as successful as the pups from the control group. The number of erroneous runs in adult rats whose mothers were subjected to hypoxia and apo-rhLF treatment during nursing did not differ from the control (one-way ANOVA with the Tukey–Kramer *post hoc* test, *q* = 3615, *p* = 0.076) or the group treated with apo-rhLF during pregnancy (one-way ANOVA with the Tukey–Kramer *post hoc* test, *q* = 2395, *p* = 0.349).

These data obtained both by the NOR and the two-level, eight-arm radial maze tests suggest that the *i.p.* administration of apo-rhLF either to pregnant or lactating rats subjected to hypoxia during pregnancy has beneficial effects on cognitive functions (short-term, long-term and working memory) of their offspring in postnatal life.

## 4. Discussion

Lactoferrin is an iron-binding, sialic-acid-rich milk glycoprotein with multifunctional health benefits important for the development of cognitive functions during the period of rapid brain growth [20]. In preclinical models of antenatal stress and perinatal brain injury, bovine LF was shown to protect the developing brain from neuronal loss, improved neuronal connectivity, increased levels of neurotrophic factors and decreased inflammation in the neuronal tissue [34,35].

Our previous studies provided evidence that apo-LF rescues HIFs from hydroxylation by prolyl-hydroxylases followed by ubiquitination [21,24]. At the same time, another research group showed that after treatment of cultured cells with LF, the hydroxylated HIF-1α was revealed by specific antibodies, which evidenced the capacity of LF to inhibit iron-dependent hydroxylases of HIF [36]. Importantly, iron-saturated LF cannot provide the same or similar effect [21,22,24]. On the contrary, it is likely to cause adverse results, probably by increasing cellular iron content, which enhances the production of ROS and launches ferroptosis [37]. Protected from hydroxylation by apo-LF, HIF-1α and HIF-2α, which are potent transcription factors, stimulate expression of multiple genes, resulting in the synthesis of many proteins, including EPO. In recent years, the evidence that EPO is able to oppose or mitigate neural lesions has been accumulating [24,38,39,40,41,42,43]. Hence, the neuroprotective effect of apo-LF is likely to occur largely due to its capacity for inducing the synthesis of EPO via a HIF-signaling mechanism.

Rats belong to a limited number of mammals that have no LF in their milk [44]. This makes them a convenient object for investigating the functions of that protein, since no accessory effect of intrinsic LF is expected [45]. Our data demonstrate that the injection of apo-rhLF in the CAPRABEL formula to pregnant rats induces the appearance of HIFs and EPO in their brain, as well as in the brain and torsos of their embryos. Moreover, in lactating dams, apo-rhLF administration leads to the appearance of rhLF in their milk and in the gastric content of their suckling pups, resulting in the upregulation of HIFs and EPO in their organs.

In this study, we have demonstrated that the apo-form of rhLF is able to improve the impaired STM and LTM, as well as working memory in the offspring of rats exposed to hypoxia during gestation. In fact, injections of apo-rhLF to both pregnant rats and to lactating dams caused a beneficial effect on cognitive functions of their offspring both at the end of the first month of development (P22) and in adulthood (P90). This is in line with the data on the corrective effect of bovine LF on the brain functions in neonate rats subjected to ischemic hypoxia [46].

The detection of HIFs and EPO in the brain and some other tissues of suckling pups, whose mothers received injections of apo-rhLF, allows us to suggest its possible role in protecting the developing brain against the harmful effects of prenatal hypoxia. This observation supports our recent finding that apo-rhLF significantly mitigates neurological symptoms in rotenone-treated rats (a model of Parkinson’s disease) and is able to rescue rats with experimental allergic encephalomyelitis (a model of multiple sclerosis) from the lethal outcome [24]. The same study showed that an *i.p.* injection of human apo-LF to mice 1 h after they were subjected to the occlusion of the medial cerebral artery significantly diminished the necrosis area in the brain [24]. A beneficial effect of bovine LF during pregnancy has also been demonstrated in the model of restricted intrauterine development [47]. In a recent study, lactating rats received fodder supplemented with bovine LF, which significantly diminished the severe brain damage caused by hypoxia-ischaemia after occlusion of their pups’ common carotid artery [48]. In all these experimental models, the neural lesions are mediated via hypoxic and oxidative stress. No wonder that triggering the potent anti-hypoxic (and anti-oxidative) pathway, by rescuing HIFs from destruction with subsequent elevated synthesis of EPO, alleviates or abrogates neurological impairments which might result in cognitive dysfunctions. The capacity of stabilized HIF to improve memory by upregulating EPO was shown previously in experiments in healthy mice [49]. The results of the present study suggest that a similar molecular mechanism might also underlie the protective effect of apo-LF against the brain damage caused by prenatal hypoxia. 

One of the possible pathways for neuroprotective effects of apo-LF observed in this study might be related to activation of cellular cascades caused by the translocation of the nuclear factor-erythroid 2 p45-related factor 2 (Nrf2) into the cell nucleus, which, as shown previously, is induced by apo-LF [24]. This effect mimics the reaction of the complex formed by Nrf2 and the Kelch-like ECH-associated protein 1 (Keap1) to oxidative stress. Under such conditions, Keap1 dissociates from Nrf2 and allows the latter to make its way to the nucleus, where it regulates the expression of numerous antioxidant stress genes [50]. This might activate neuroprotective mechanisms and protect neuronal cells and cognitive functions of animals as observed in this study through other anti-inflammatory mechanisms in which LF might be involved [51].

## 5. Conclusions 

It can be concluded that the capacity of lactoferrin in its iron-free apo-form to rescue HIFs from destruction and, as such, to upregulate EPO and other target gene expression alleviates the detrimental effects of prenatal hypoxia on the brain of developing animals, both when apo-rhLF in the CAPRABEL formula is administered in pregnancy or lactation. As a result, apo-rhLF administration improves the cognitive functions of the offspring subjected to prenatal hypoxia and can be recommended for designing therapeutic strategies to reduce cognitive deficits caused by pathological pregnancy or labor.

## Figures and Tables

**Figure 1 nutrients-14-01399-f001:**
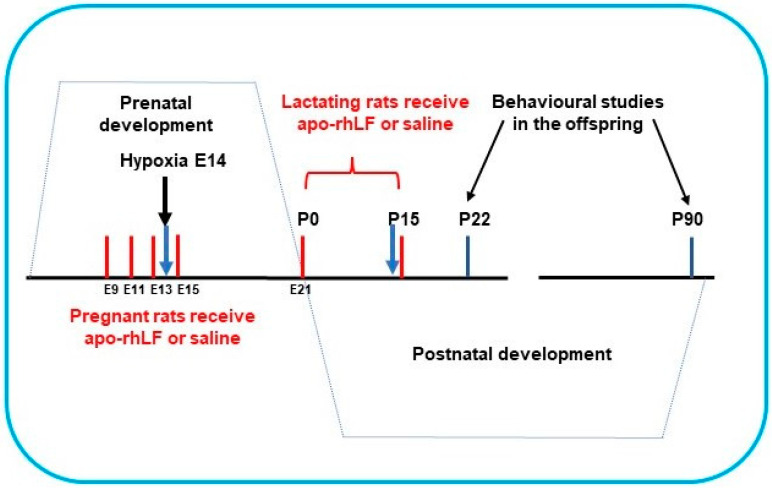
The scheme of apo-rhLF administration to pregnant and lactating rats. Red lines—days of apo-rhLF administration to pregnant or lactating rats. Blue arrows mark the periods of collecting the material for Western blotting (E14 and P14).

**Figure 2 nutrients-14-01399-f002:**
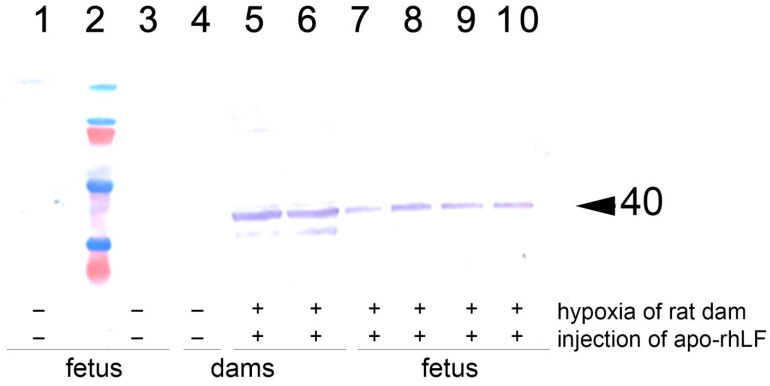
Western blotting analysis with anti-EPO of brain homogenates from rat females and their embryos. 1 and 3: Brain samples of the embryos from rats subjected to hypoxia without injection of apo-rhLF; 4: brain sample of an intact pregnant rat injected with saline (no hypoxia); 5 and 6: brain samples of the embryos from rats injected with apo-rhLF during pregnancy; 7–10: brain samples of the embryos from rats injected with apo-rhLF; 2: molecular mass markers; arrow marks M 40 kDa. Hypoxia −/+ or apo-rhLF −/+ indicate whether hypoxia or apo-rhLF have been applied.

**Figure 3 nutrients-14-01399-f003:**
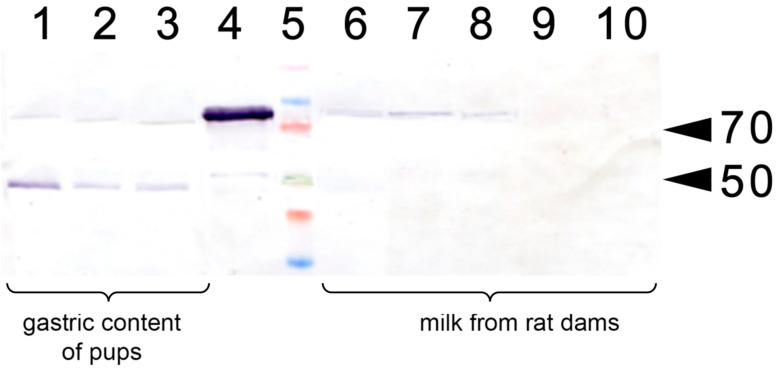
Western blotting with anti-LF antibody of lactating rat milk and of the gastric content of their pups. 1–3: Gastric content of pups (P14) fed by the rats injected with apo-rhLF; 4: control sample of CAPRABEL (1 µg); 5: molecular mass markers, arrows mark 70 and 50 kDa; 6–8: milk from rats injected with apo-rhLF; 9–10 milk from control rats injected with saline.

**Figure 4 nutrients-14-01399-f004:**
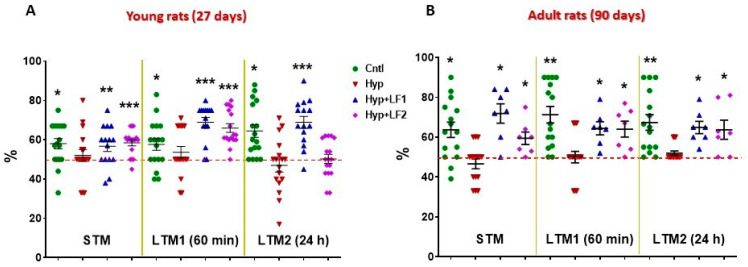
Results of testing short-term memory (STM) and long-term memory (LTM1 and LTM2) in the offspring of hypoxia-treated rats by Novel Object Recognition test. Ordinate: mean ± SEM of the time spent exploring novel objects expressed in % of the total time spent exploring both the novel and familiar objects. (**A**)—Rats tested on postnatal day 22; (**B**)—Rats tested on postnatal day 90. *—*p* ≤ 0.05; **—*p* ≤ 0.01; ***—*p* ≤ 0.01 (one sample test) differences between the time spent exploring the novel object and the hypothetical 50% value (the horizontal line) in case of equal preference for the novel and familiar objects. The number of animals in each group is specified in the Methods. Hyp—hypoxia; Hyp+LF1—hypoxia and apo-rhLF during gestation; Hyp+LF2—hypoxia and apo-rhLF during lactation.

**Figure 5 nutrients-14-01399-f005:**
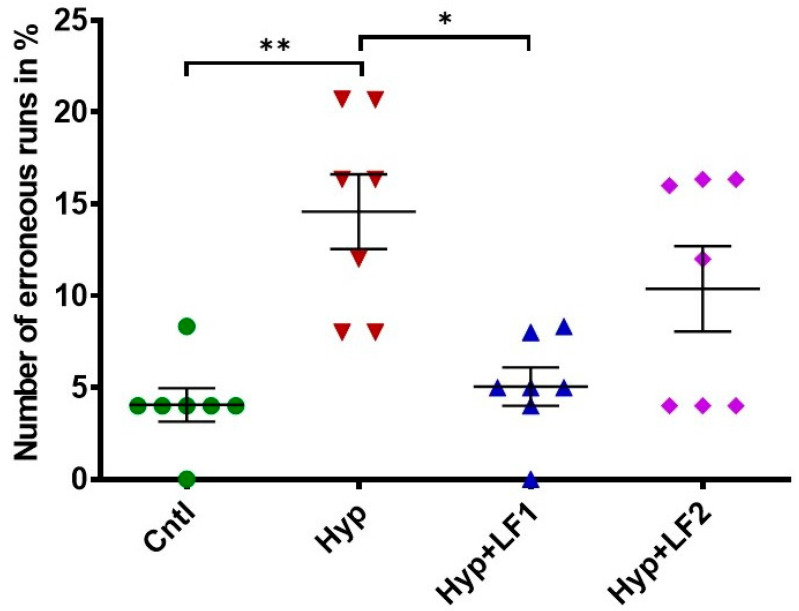
Effect of apo-rhLF on the working short-term memory of adult offspring (P90) tested in the two-level, eight-arm radial maze. Ordinate: number of erroneous runs in % (mean ± SEM). *—*p* ≤ 0.05; **—*p* ≤ 0.01—difference between Hyp and other groups (one-way ANOVA: F_3,27_ = 8.064, *p* = 0.001, Tukey–Kramer *post hoc* test). The number of animals in each group is specified in the Methods. Hyp—hypoxia; Hyp+LF1—hypoxia and apo-rhLF during gestation; Hyp+LF2—hypoxia and apo-rhLF during lactation.

**Table 1 nutrients-14-01399-t001:** Groups of experimental animals and the means of their analysis.

Rat Groups (N)	Substance Injected	Biochemical Tests	Behavioral Tests
Control with saline	Saline (0.5 mL; E9, 11, 13, 15)	2 females and 10 embryos (E14); WB-EPO	NOR (P22, *n* = 17 and P90, *n* = 15) 8 arm-maze, (P90, *n* = 7)
Hypoxia (E14) with saline	Saline (0.5 mL; E9, 11, 13, 15)	2 females and 16 embryos (E14); WB-EPO 4 females and 11 pups (P14); WB: EPO, HIF1α and HIF2α	NOR (P22, *n* = 17 and P90, *n* = 15) 8 arm-maze, (P90, *n* = 7)
Hypoxia (E14) with apo-rhLF during gestation—Hyp+LF1	Apo-rhLF (10 mg in 0.5 mL of saline; E9, E11, E13, E15)	2 females and 12 embryos (E14); WB: EPO	NOR (P22, *n* = 15 and P90, *n* = 7) 8 arm-maze, (P90, *n* = 7)
Hypoxia (E14) with apo-rhLF during lactation—Hyp+LF2	Apo-rhLF (10 mg in 0.5 mL of saline; daily from P0 to P15 days after birth)	4 females and 16 pups (P14); WB: EPO, HIF1α and HIF2α	NOR (P22, *n* = 17 and P90, *n* = 7) 8 arm-maze, (P90, *n* = 7)
Control with apo-rhLF during lactation	Apo-rhLF (10 mg in 0.5 mL of saline; daily from P0 to P15 days after birth)	3 females and 9 pups (P14); WB: EPO, HIF1α and HIF2α;	
Control with apo-rhLF during lactation	Apo-rhLF (single injection 10 mg in 0.5 mL of saline on P14)	3 females and 6 pups (P14) WB: LF in dam milk and in the pup gastric content	

**Table 2 nutrients-14-01399-t002:** EPO detection by Western blotting in pregnant rats and their embryos.

Groups		Analyzed Part of Pregnant Rats or Fetus			
		Brain of Pregnant Females	Placenta	Brain of the Embryos	Torso of the Embryos
Embryos (*n* = 10) from control pregnant rats (*n* = 2) with *i.p.* saline injections during gestation	EPO presence	0	0	0	0
	EPO absence	2	10	10	10
Embryos (*n* = 16) from pregnant rats (*n* = 2) with hypoxia and *i.p*. saline injections during gestation	EPO presence	0	0	0	0
	EPO absence	2	16	16	16
Embryos (*n* = 12) from pregnant rats (*n* = 2) with hypoxia and *i.p.* apo-rhLF injections during gestation	EPO presence	2	0	12	12
	EPO absence	0	12	0	0

**Table 3 nutrients-14-01399-t003:** HIF-1α, HIF-2α and EPO detection by Western blotting in organs of pups.

Organ	Pups (*n* = 9) of Control Rats with *i.p.* apo-rhLF Injections during Lactation (*n* = 3)		Pups (*n* = 11)of Rats after Hypoxia and *i.p.* Saline Injections during Lactation (*n* = 4)		Pups (*n* = 16)of Rats after Hypoxia and *i.p.* apo-rhLF Injections during Lactation (*n* = 4)	
	HIF1α HIF2α and EPO Presence	HIF1α HIF2α and EPO Absence	HIF1α HIF2α and EPO Presence	HIF1α HIF2α and EPO Absence	HIF1α HIF2α and EPO Presence	HIF1α HIF2α and EPO Absence
Brain	9	0	0	11	16	0
Liver	9	0	0	11	16	0
Spleen	9	0	0	11	16	0

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
