# Peer review of "Lactoferrin Induces Erythropoietin Synthesis and Rescues Cognitive Functions in the Offspring of Rats Subjected to Prenatal Hypoxia"

_nutrients, 2022, doi:10.3390/nu14071399_

Round 1

Reviewer 1 Report

The paper is well structured and the study is well designed. It reminds me a bit of studies done on rats proving the positive effect of magnesium sulfate on cognitive functions in preterm delivery. The authors start with showing that lactoferrin which is not found in the milk of rats is detected in their milk and gastrointestinal tract after they are supplemented with Recombinant human apo-lactoferrin (rhLF). After inducing hypoxia they show that rhLF causes the up regulation of EPO. This in turn (and here I would be a bit careful) protects against brain damage. 

I would only probably add a sentence before the last sentence:

 "As a result, apo-rhLF administration improves the cognitive functions of the offspring subjected to prenatal hypoxia and can be recommended for designing therapeutic strategies to reduce cognitive deficits caused by pathological pregnancy or labour."

and I would say, more studies or studies on a larger number of.....is required to prove the point. 

I think this is a very good study and probably there is a chance for rhLF to be used therapeutically in the future but is it really enough to conclude that from such a small number of rats studied?

The English of the paper is very good and does not need any corrections. 

Author Response

Many thanks for positive evaluation of our manuscript. We agree that it would be important to extend our study using a larger number of the offspring in any future experiments. In the current study, the statistical analysis confirmed significant difference for the number of animals used.

Reviewer 2 Report

This study of Sokolov et al. aims to test the neuroprotective effect of Lactoferrin injection (Lf) into female rates during gestation or breastfeeding, in the context of preterm hypoxia-ischemia. Erythropoietin as well as HIFs protein levels were evaluated by western blots in divers dam, fetus and pup tissues. Functional neuroprotection was assessed using the novel object recognition test (on short and long-term memory).

To improve this article, there are a number of points that should be corrected.

Major points:

Point 1: Experimental design.

Experimental design looks scientifically curious at different levels. Firstly, the intraperitoneal mode of administration has a pro-inflammatory character. The authors could have administered lactoferrin via maternal supplementation (as other studies have done, for e.g. Sanchez et al. Nutrients 2021). This less invasive approach is more conducive to future translation to humans. Second, injections were given every 2 days for pregnant females and daily for lactating females. Why such a difference in the protocol? Similarly, for the group of pregnant females, the injections were performed within a time window of 7 days, while for the group of lactating females, the injections were performed during a period of 15 days. Same question, why such a difference in the protocol?

Point 2: Number of pups.

There is great heterogeneity in the groups (Table 1). For example, why are there so many females in (n = 4) in the “Hypoxia with apo-rhLF during lactation” group? Why are there so few pups in the Nor P90 “Hypoxia with apo-rhLF during gestation” group (n = 5)? Moreover, such a small number of pups does not seem sufficient to have robust data in a behavioral test such as NOR.

Point 3: Table 3.

The titles of the columns of table 3 are very difficult to read, separations should be added or the table should be redrawn.

Point 4: Figure 5.

Results are reported as mean +/- sem. Why are error bars not visible? In the legend, it is indicated that green corresponds to the Ct group; however no group appears in green in the graphic.

Point 5: Stat plots.

When it is possible, individual data should be plotted in histograms (stat plots).

Point 6: lines 281-290.

The authors' data do not make it possible to highlight these signaling pathways. That part of the discussion should be removed.

Point 7:

The terms embryo and fetus are not equivalent. Authors should modify and standardize these terms for clarity.

Minor points:

Point 1: Spaces between numbers and units must be homogenized, as must abbreviations.

Point 2: Thank you for agreeing to homogenize the injection times:  line 107 “prenatal hypoxia 106 with apo-rhLF treatment during nurturing P1-15” vs line 112 “daily after 111 delivery on P0 up to P15”

Author Response

Point 1. 

The intraperitoneal mode of hrLF administration to animals have been widely used in many studies some of which demonstrating its anti-inflammatory effects (Hi ei al, J Dairy Sci, 2021 104(7):7383-7392, doi: 10.3168/jds.2020-19232). We agree that oral administration would be more physiological but will require larger amounts of hrLF to be consumed and the equality of the amount of the substance absorbed from the digestive system to the maternal organism would be difficult to estimate. Moreover, the major objects of our study were embryos and offspring of rats to whom hrLF was delivered either via placenta or milk.

The difference in the protocol of hrLF administration to the pregnant and lactating dams was adjusted to the timescale of brain development in the embryos and offspring. Fewer injections to the pregnant rats were considered to induce less stress which also can affect brain development of the embryos. Daily administration of hrLF to lactating rats was dictated by the fact that according to our preliminary data and the data presented in this paper stable presence of hrLF in the milk of lactating rats was detected only during 4-24 h after its i.p.administration.

Point 2. The number of animals in each group was dictated by the necessity to obtain sufficient amount ofmaterial for the analysis (including milk). The number of animals tested in the “Hypoxia with apo-rhLF during gestation” was erroneously stated as 5. In fact, there was an equal number of animals tested (n=7) which is now corrected in the text and reflected in dot-plot figures. As we have already written in response to the reviewer 1, in the current study, the statistical analysis confirmed significant difference for the number of animals used.

Point 3.

The titles of the columns have been edited and justified for easier reading.

Point 4.

The figures 4 and 5 have now been presented as dot plots and all necessary indications are given in the graphs.

Point 5.

Done. See above.

Point 6.

Indeed, the data obtained in this study have no direct relation to the signaling pathways mentioned in Discussion. However, the passage cannot be removed, since it meets the requirements of another Reviewer.

Point 7.

Many thanks for this comment. The term “fetuses” in the manuscript has been corrected for “embryos”.

Minor.

Point 1.

This error occurred most probably due to minor differences among the version of the Microsoft text editor. We shall control the final version of the manuscript.

Point 2.

This has been unified.

Reviewer 3 Report

The article investigated the protective effect of apo human recombinant lactoferrin (apo-rhLF) on prenatal hypoxia induced cognitive function impairment. Apo-rhLF i.p. injection ameliorated the cognitive deficits caused by prenatal hypoxia and improve the memory in young and adult offspring using radial maze and recognition test. The authors proposed the protective effect of apo-rhLF on brain development was via a lactoferrin induced HIF-EPO signaling pathway. Despite of the improvements in the phenotypes of mice receiving apo-rhLF administration, there was flaws in the proposed mechanisms. In addition, experiments in the manuscript are not persuasive and additional assays need performing. If the mechanistic part is not solid, the phenotype part of the manuscript is only reinforcing the efficacy of lactoferrin in the protection of rat brain injury from hypoxia. Please refer to a recent published article: Eduardo Sanches et al, dose-dependent neuroprotective effects of bovine lactoferrin following neonatal hypoxia-ischemia in the immature rat brain. Nutrients 2021.

Specific comments

  1. Author regarded “the neuroprotective effect of apo-LF is likely to occur largely due to its capacity for inducing the synthesis of EPO via a HIF-signaling mechanism”, and the EPO levels in neonatal brain homogenates were detected. However, the detection of HIF signaling in same samples is lacking (Figure 2), although the author claimed that “our studies demonstrated that various tissues of animals treated with recombinant 72 human LF (rhLF) responded by expressing HIF-1α and HIF-2α target genes”. If the proposed mechanism is valid, HIF signaling as well as downstream genes in the pup’s brain receiving apo-rhLF are expected to be upregulated when compared with controls receiving hypoxia alone.

  1. If the author hypothesized apo-rhLF protects cognitive function by up regulation of EPO, comparison between mice receiving hypoxia treatment alone and those receiving hypoxia treatment + apo-rhLF should be performed. This comparison is lacking in the blot (Figure 2). Hypoxia alone can induce the expression of EPO, whether apo-rhLF can further enhance the EPO expression under hypoxia is unknown. Based on current data, authors cannot conclude that apo-LF induced the synthesis of EPO under hypoxia.

  1. In a recent article by Mari Ibuki et al ( Pharmacol. 2020;11:174), they found that HIF-1α protein was suppressed by the administration of lactoferrin in mouse retinal neuronal cells. It contradicts with the proposed mechanism in this study. How to explain the difference between studies?

  1. In Table 2, lack of treatment group (embryos from control pregnant rats with i.p. apo-rhLF in jections during gestation) compare with control pregnant rats receiving saline. How to explain the total absence of EPO in embryos from rat under hypoxia in Table 2?

  1. In Table 3, there is no control group (pups from control rats with i.p. saline injections during lactation) for “pups (n=9) from control rats with i.p. apo-rhLF injections during lactation (n=3)”.

  1. Representative gel blot images are recommended to be presented in the manuscript (Table 2 and 3).

  1. Lactoferrin possesses iron-chelation effect, which confers lactoferrin antioxidant and anti-inflammatory properties. Hypoxia-induced inflammation is regarded to contribute to the cognitive impairment. Is it possible the anti-inflammatory effect of apo-rhLF rescue the cognitive performance? Please include relevant discussion in the manuscript.

  1. For figure 4 and 5, scatter plot with error bars is recommended. Please indicate n number in the figure or figure legend instead of in the main text. Please label P values in the figure.

Author Response

Responses to the specific comments.

  1. In our detailed previous study (Zakharova et al., 2018; DOI: 10.1007/s10534-018-0111-9) we showed a definite response of various animal tissues to the treatment with rhLF by stabilizing HIF-1a and increased synthesis of EPO. Therefore, in this study we focused on the effect of prenatal hypoxia on this mechanism. This was our specific goaland repeating previous experiments would distract the reader from the mainstream of the study.
  2. Our experiments were in process when Zhang et al., 2021 (https://doi: 10.7150/thno.52028 ) published their paper showing that hypoxia and lactoferrin in combination gave an effect more pronounced than the “arithmetic sum” of the two separate effects. An appropriate reference can be found in our manuscript. Using antagonist(s) of the EPO receptor might be a good instrument for testing the causative link between EPO synthesis and the cognitive functions of animals. This will probably be the next step of our research.
  3. Along with the paper proposed by the Reviewer, there are quite a few studies documenting the stabilization of HIF-1a caused by lactoferrin. The paper by Mari Ibuki et al. does not propose any molecular mechanism of lactoferrin action that might contradict to the one proposed in our study. In fact, that paper describes a mere phenomenon of HIF-1a suppression in a murine model of choroidal neovascularization. Since the authors did not specify which type of lactoferrin was used in their study, the holo-protein seems a likely species. Iron-containing lactoferrin does not rescue HIF-1a from degradation by iron-dependent prolyl hydroxylases, contrary to the effect of apo-protein used in our study. As shown in the paper by Zhang et al., 2021 (https://doi: 10.7150/thno.52028) cited in our manuscript, holo-lactoferrin increases HIF-1a degradation under hypoxia. Because the eye tissues are highly penetrant to oxygen, the results obtained in the model used by Ibuki et al. are not easily comparable with those obtained for other organs or tissues of a mammalian organism.
  4. Since we previously showed that administration of lactoferrin caused EPO synthesis via HIF-1a signaling, the Ethical Committee did not approve the inclusion of a separate group of animals in the protocol, because the forthcoming result for that group was obvious. Rat embryos are normally developing in hypoxic conditions. It seems likely that in the case of EPO responsive synthesis in their bodies no negative effect of external hypoxia would occur and not a single experiment like in our study could give definite results.
  5. Similarly, the Ethical Committee did not approve the presence of such an animal group.
  6. In accord with the recommendations, we supplied Table 2 with respective gel blot images and placed those in the Supplementary materials.
  7. We are grateful to the Reviewer for this suggestion and appropriatecomment was included in the text. A short discussion of the anti-inflammatory activity of lactoferrin is discussed in this manuscript on the basis of the protein’s capacity to activate Nrf2 transcription factor, described in our previous paper (Zakharova et al., 2018; DOI: 10.1007/s10534-018-0111-9). In accord with the Reviewer’s suggestion we added a reference to a comprehensive review by Lepanto et al., 2019; (DOI:10.3390/molecules24071323) concerning the anti-inflammatory effect of lactoferrin. Our research was not dedicated to studying anti-inflammatory effects of lactoferrin, yet this aspect deserves further exploration.
  8. The figures 4 and 5 have now been presented as dot plots and all necessary indications are given in the graphs.

Round 2

Reviewer 2 Report

Authors have correctly responded to comments.

Reviewer 3 Report

None